# The NMD Pathway Regulates *GABARAPL1* mRNA during the EMT

**DOI:** 10.3390/biomedicines9101302

**Published:** 2021-09-23

**Authors:** Timothée Baudu, Chloé Parratte, Valérie Perez, Marie Ancion, Stefania Millevoi, Eric Hervouet, Anne Peigney, Paul Peixoto, Alexis Overs, Michael Herfs, Annick Fraichard, Michaël Guittaut, Aurélie Baguet

**Affiliations:** 1INSERM, EFS BFC, UMR1098, RIGHT Institute, Interactions Hôte-Greffon-Tumeur/Ingénierie Cellulaire et Génique, University Bourgogne Franche-Comté, 25000 Besançon, France; timothee.baudu@edu.univ-fcomte.fr (T.B.); chloe.parratte@edu.univ-fcomte.fr (C.P.); valerie.perez@univ-fcomte.fr (V.P.); eric.hervouet@univ-fcomte.fr (E.H.); anne.peigney@univ-fcomte.fr (A.P.); paul.peixoto@univ-fcomte.fr (P.P.); alexis.overs@edu.univ-fcomte.fr (A.O.); annick.fraichard@univ-fcomte.fr (A.F.); michael.guittaut@univ-fcomte.fr (M.G.); 2Laboratory of Experimental Pathology, GIGA-Cancer, University of Liege, B-4000 Liege, Belgium; marie.ancion@uliege.be (M.A.); m.herfs@uliege.be (M.H.); 3Cancer Research Centre of Toulouse, INSERM UMR 1037, Université Toulouse III-Paul Sabatier, 31330 Toulouse, France; stefania.millevoi@inserm.fr; 4DImaCell platform, University Bourgogne Franche-Comté, 25000 Besançon, France; 5EPIGENEXP platform, University Bourgogne Franche-Comté, 25000 Besançon, France; 6Laboratoire de Biochimie, CHU Besançon, 25000 Besançon, France

**Keywords:** NMD, autophagy, ATG8, *GABARAPL1*, *GABARAP*, *LC3*, lung cancer

## Abstract

EMT is a reversible cellular process that is linked to gene expression reprogramming, which allows for epithelial cells to undergo a phenotypic switch to acquire mesenchymal properties. EMT is associated with cancer progression and cancer therapeutic resistance and it is known that, during the EMT, many stress response pathways, such as autophagy and NMD, are dysregulated. Therefore, our goal was to study the regulation of ATG8 family members (*GABARAP, GABARAPL1*, *LC3B*) by the NMD and to identify molecular links between these two cellular processes that are involved in tumor development and metastasis formation. IHC experiments, which were conducted in a cohort of patients presenting lung adenocarcinomas, showed high *GABARAPL1* and low *UPF1* levels in EMT+ tumors. We observed increased levels of *GABARAPL1* correlated with decreased levels of NMD factors in A549 cells in vitro. We then confirmed that *GABARAPL1* mRNA was indeed targeted by the NMD in a *3′UTR*-dependent manner and we identified four overlapping binding sites for *UPF1* and *eIF4A3* that are potentially involved in the recognition of this transcript by the NMD pathway. Our study suggests that *3′UTR*-dependent NMD might be an important mechanism that is involved in the induction of autophagy and could represent a promising target in the development of new anti-cancer therapies.

## 1. Introduction

The epithelial-to-mesenchymal transition (EMT) is a physiological process in which epithelial cells lose their cell polarity and acquire mesenchymal properties; it occurs during embryogenesis or tissue remodeling (EMT I and II). During cancer progression, the EMT (type III) [1] plays a key role in metastasis formation by promoting cellular invasion, dissemination and tissue colonization. Specific morphological changes observed in cells undergoing the EMT are controlled at different levels of gene expression, including epigenetics, transcriptional and post-transcriptional levels, or translation [2]. It was reported that the EMT has an impact on autophagy, and conversely, the induction of autophagy inhibits the EMT by regulating gene expression at multiple levels from epigenetic changes to mRNA processing. During the EMT, many cellular pathways were described as being altered or dysregulated, especially the ones linked to the stress response. One of them is autophagy, which was extensively described to be activated during the EMT to maintain cell homeostasis and cell survival. To do so, autophagy can induce the degradation and recycling of intracellular components, such as protein aggregates or damaged organelles. The autophagy process is mediated by more than 30 autophagy-related (ATG) proteins and, amongst these proteins involved in autophagy, two subfamilies were described as the homologs of the yeast Atg8 (autophagy-related 8) in mammals: the MAP-LC3s (microtubule-associated protein light chain 3) comprising *LC3A*, *LC3B* and *LC3C*, and the *GABARAP* (GABA_A_-receptor-associated protein) family. The latter also includes three members: *GABARAP*, *GABARAPL1*/GEC1 (*GABARAP*-like protein 1/guinea-pig endometrial glandular epithelial cells-1) and *GABARAPL2*/GATE-16 (*GABARAP*-like protein 2/Golgi-associated ATPase enhancer of 16 kDa). The human *GABARAP*, *GABARAPL1* and *GABARAPL2* genes are located on chromosomes 17p13.12, 12p12.3 and 16q22.3, respectively, and were described to be differentially expressed in normal and pathological tissues [3]. Even if the function of these proteins during autophagosome formation and elongation is now widely admitted, the regulation of their expression is still poorly understood. Indeed, these proteins were mainly studied at the protein level and limited data describe the regulation of their transcription or post-transcription regulation in response to stress induction. Amongst these studies, the regulation of autophagy genes was focused on their transcriptional control. For example, the transcription factor ATF4 was shown to activate the transcription of several autophagy genes, including *ATG5* and *ATG8* [4,5]. Therefore, in our study, we decided to investigate the post-transcriptional regulation of mRNAs encoding ATG8 proteins by the NMD, namely, nonsense-mediated mRNA decay.

NMD is an evolutionary conserved cellular process that is present in all eukaryotes; it controls the quality of mRNAs via the selective targeting and degradation of premature termination codon (PTC)-harboring mRNAs. It was first described in the etiology of β-thalassemia [6,7] in which the presence of the NS39 nonsense mutation (NS) in the second exon of the β-globin gene leads to the appearance of a PTC-carrying mRNA that is recognized and targeted for degradation by the NMD. The elimination of the mRNA ultimately leads to the absence of β-globin protein synthesis and the development of disease. Therefore, NMD represents an essential post-transcriptional regulatory mechanism that limits the synthesis of truncated proteins. At the molecular level, NMD takes place during the first round of translation through the recruitment of UPF (up-frameshifting factors, including *UPF1*, 2, 3A and 3B, where the latter is also named UPF3X) and SMG (suppressor of morphogenetic effect on genitalia proteins, including SMG1, 5, 6, 7 and 8) factors. The activation of the NMD occurs via two distinct pathways: an exon junction complex (EJC)-dependent NMD, which is the main “canonical” pathway of activation, and an EJC-independent NMD. Nevertheless, both these ways require *UPF1*, which is the core factor and the main regulator of the NMD [8]. The EJC is a multiprotein complex that plays a key role in the EJC-dependent NMD by discriminating a PTC from a physiological termination codon. The EJC is normally deposited 20–24 nucleotides upstream from the exon–exon junctions during splicing [9,10,11]. This complex is composed of a core of four proteins: *eIF4A3*, a DEAD-box RNA helicase family member, MLN51 and a heterodimer formed by the association of MAGOH and Y14 proteins. Thus, EJC plays a major role in RNA processing, including the NMD pathway, by promoting the recruitment of NMD factors to elicit the degradation of PTC-containing transcripts [12,13,14]. More recently, another model of NMD that is independent of the EJC was also reported [15,16]. In the EJC-independent NMD model, the PTC recognition does not happen through the presence on the transcript of an EJC but is linked to the *3′UTR* length. NMD is indeed able to target transcripts harboring NMD-inducing features, such as a long 3′-UTR (>1000 nt), leading to the conclusion that about 10% of physiological transcripts could be targeted by the NMD [17]. Beyond its role in surveillance and degradation of aberrant transcripts harboring a PTC, it has also been shown that NMD can act as a post-transcriptional regulatory mechanism targeting physiological transcripts.

In this study, we demonstrated that the *GABARAPL1* mRNA was regulated at the post-transcriptional level by the NMD during the EMT. According to our results, the mechanism by which NMD targeted *GABARAPL1* mRNA occurred through the association of both *UPF1* and *eIF4A3* on its *3′UTR*. These data indicated that during the EMT, NMD inhibition would lead to an increase of *GABARAPL1* levels, suggesting a role for the NMD pathway in the regulation of autophagy.

## 2. Materials and Methods

### 2.1. Antibodies, Vectors, Chemicals and siRNAs

For the Western blotting experiments and the IHC, the following antibodies were used: rabbit polyclonal anti-*GABARAPL1* (D5R9Y, Cell Signaling, Danvers, MA, USA), rabbit polyclonal anti-*UPF1* (D15G6, Cell Signaling), rabbit polyclonal antibody anti-SMG1 (Q25, Cell Signaling), rabbit polyclonal anti-b-actin (A5060, Sigma Aldrich, St. Louis, MI, USA), monoclonal anti-Vimentin (#7902917, Ventana Medical Systems, Oro Valley, AZ, USA), secondary goat polyclonal anti-rabbit HRP (BI2407, Abliance, Compiègne, France) and anti-mouse HRP (BI2413C, Abliance).

For the cellular treatments, cycloheximide (01810, Sigma, St. Louis, MI, USA), G418 (11811023, ThermoFisher, Santa Cruz, CA, USA) and NMDi14 (SML1538, Sigma-Aldrich) were used.

The *pGFP*-*GABARAP* vector cloning was previously reported in [18]. The *pGFP*-*GABARAP* + *3′UTR*-*GABARAPL1* was synthesized by amplifying the *3′UTR* of *GABARAPL1* added with SacII and BamHI restriction sites via PCR using the following primers: forward—5′-TAAGCACCGCGGGTGGTTGGAAGCCCAGCAG and reverse—ATTTTATTAACGAATGGAATTTTTAGCCTAGGATTCGT. The amplified sequence was then cloned downstream *GABARAP* CDS in a *pGFP*-*GABARAP* plasmid using the SacII and BamHI sites.

The siRNAs used in the transfection were purchased from Eurogentec (Seraing, Belgium). The sequences are listed in Table 1.

### 2.2. Cell Culture and Transfection

The A549 (NSCLC) cell line was obtained from Dr. Christophe Borg (INSERM UMR1098, Besançon, France) and the SiHa cells were obtained from ATCC (HTB-35™). A549 cells were grown in Dulbecco’s minimum essential medium (DMEM) 1 g/L glucose (L0066, Dominique Dutscher, Brumath, France) and SiHa cells were cultured in RPMI1640 medium (Dutscher, L0498). Both media were supplemented with fetal bovine serum (10%) (S1810, Dominique Dutscher). Cells were cultured at 37 °C in 5% CO_2_ and routinely used at 70–80% confluence.

Transfections of the plasmids were performed using the Lipofectamine 2000 reagent (11668030, ThermoFisher) following the supplier’s recommendations. Briefly, cells were seeded in 6-well plates and transfected for 24 h with 1 µg of plasmid and 2 µL of Lipofectamine 2000 reagent and harvested 24 h after the end of the transfection. Transfections of the siRNA were conducted using the same protocol with 20 pmol of siRNA and 3 µL of Lipofectamine 200 reagent.

### 2.3. Western Blotting

For the total protein extracts, cells were scraped, harvested and lysed in lysis buffer (10 mM Tris, 1 mM EDTA, 1 mM PMSF, 1 mM NA-Vanadate, 1% DOCA) supplemented with protease inhibitors (104 mM AEBSF, 1.5 mM pepstatin A, 1.4 mM E-64, 4 mM bestatin, 2 mM leupeptin, 80 µM aprotinin) for 30 min on ice and sonicated for 15 s (Sonics and Materials, Vibra-Cell, Newtown, CT, USA). Following a 5 min incubation at 95 °C, protein lysates were then separated on TGX acrylamide gels (1610172, Biorad, Hercules, CA, USA) at 150 V for 45 min using the Protean 3 system before being transferred onto Transblot turbo PVDF (1704157, Bio-Rad) membranes in a transfer buffer (Bio-Rad, 1704273) for 7–10 min using the Transblot turbo system (1704150, Biorad) according to the manufacturer’s recommendations. Membranes were then saturated in 5% non-fat milk or BSA diluted in Tris-buffered saline supplemented with Tween 20 (TBS-T) (20 mM Tris base, pH 7.6, 137 mM NaCl, 0.1% Tween 20) for 1 h and incubated with primary antibodies, according to the manufacturer’s recommendations, in TBS-T supplemented with 5% non-fat milk or BSA overnight at 4 °C. Following three washes in TBS-T, immunoreactive bands were detected using a secondary goat horseradish peroxidase (HRP)-coupled secondary anti-rabbit antibody (1/10,000, BI2407, Abliance) or HRP-coupled secondary anti-mouse antibody (1/10,000, BI2413C, Abliance) and the Clarity Western ECL blotting substrate (Biorad, 1705061). The signals were then analyzed and quantified using the ChemiDoc TMXRS+ system (1705051, Biorad) and the Image Lab software (Biorad). Normalization was done using the Stain-Free technology (Biorad).

### 2.4. RNA Extraction and RT-qPCR

The total RNAs were extracted using the Tri Reagent (Molecular Research Center, TR-118, Cincinnati, OH, USA) following the supplier’s recommendations. Briefly, the cell pellet (300,000 cells) was suspended in 750 µL of Tri Reagent and 0.2 mL of chloroform was added before centrifugation at 12,000× *g* for 15 min at 4 °C. The total RNAs were then precipitated using 0.5 mL of isopropanol and centrifuged at 10,000× *g* for 20 min at 4 °C. The RNA pellet was washed with 70% ethanol before solubilization in water (30 µL) and then incubated for 10 min at 65 °C. Total purified RNAs were then quantified before conservation at −80 °C.

For RT-qPCR analysis, 1 µg of total RNAs were reverse-transcribed using 60 units of M-MLV Reverse Transcriptase (Sigma, M1302), 1.25 µM of oligo (dT)23 (Eurogentec) and 1.25 µM of random primers (Promega, C1181, Madison, WI, USA). Quantitative PCR was run on the Step One Real-Time PCR System (Applied Biosystems, Waltham, MA, USA) using the Syber Green PCR Master Mix (Applied Biosystems, 4309155) and the following parameters: 10 min at 95 °C followed by 40 cycles of 15 s at 95 °C and 1 min at 60 °C. The target gene levels (see the primers in Table 2) were normalized to the ratio of 2 housekeeping gene expressions: *H3B2* and *18S* rRNA. The primer sequences used for qPCR analysis are described in Table 2. Each sample was analyzed in duplicate and then differences in the expression of each gene were quantified using the ΔΔCT method.

### 2.5. RNA-FISH

The cells were fixed in 4% paraformaldehyde for 30 min at room temperature and RNA-FISH assays were performed using the RNAscope^®^ Fluorescent Multiplex Assay (Advanced Cell Diagnostics Bio, Milan, Italy) following the supplier’s recommendations. Briefly, cells were dehydrated with increasing concentrations of EtOH (50, 70 and 100% for 2 min) and rehydrated with decreasing concentrations of EtOH (70 and 50% for 2 min). Cells were then incubated with Protease III for 10 min at room temperature and subjected to *GABARAP* and *GABARAPL1* probe hybridization and counterstaining with DAPI. Slides were mounted in Vectashield (Polysciences Inc., Warrington, PA, USA) and observed using Z-section imaging with a confocal microscope (Zeiss LSM 800 AiryScan, Oberkochen, Germany).

### 2.6. Immunohistochemistry and Immunostaining Assessment

FFPE specimens of NSCLCs were collected (in collaboration with the Tissue Biobank of the University of Liege, Liege, Belgium) and stained with the antibodies listed in Appendix A. The protocol was approved by the Ethics Committee of the University Hospital of Liege and the diagnosis of each case was confirmed by experienced histopathologists. IHC analysis was performed using a standard protocol detailed previously [19]. Samples were classified into two groups, namely, EMT-positive vs. EMT-negative, depending on their Vimentin immunoreactivity. As previously described [19,20], the IHCs were scored by multiplying two values: intensity (0–3) and extent (0–3), leading to a global score ranging between 0 and 9. Using this latter arbitrary scale, all immunolabelled tissues were evaluated independently by two experienced histopathologists.

### 2.7. RNA-Immunoprecipitation (RIP)

A549 cells (10 millions) cultured in 10 cm Petri dishes were scrapped and harvested in a Lysis buffer (50 mM Tris, pH 7.4, 150 mM NaCl, 10 mM MgOAc, 0.05% NP-40, 1 mM DTT, 1× protease inhibitors w/o EDTA, 200 U/mL RNaseOUT). Total cellular RNAs bound to *UPF1* were then immunoprecipitated (IP) using an anti-*UPF1* antibody (D15G6, Cell Signaling) and Dynabeads Protein A magnetic beads (53033, Active Motif, Carlsbad, CA, USA). The beads were first complexed to the antibody for 1 h at 4 °C and then IPs were conducted for 6 h at 4 °C using the IP-Star Compact Automated System (B03000002, Diagenode, Denville, NJ, USA).

Following extensive washing in a Lysis buffer, immunoprecipitated mRNAs and proteins were analyzed using RT-qPCR and Western blotting, respectively. To do so, bound complexes were eluted using the Elution buffer (100 mM Tris-HCl, pH 6.8, 20% glycerol, 4% SDS, 12% β-MeSH). Immunoprecipitated mRNAs were then purified using the Tri Reagent extraction protocol described before and proteins were denatured for 5 min at 95 °C prior to the Western blot analysis.

### 2.8. Statistical Analysis

Statistical analyses were performed using Student’s *t*-test. Data are expressed as the mean ± S.E.M. ns: Not significant, *: *p* ≤ 0.05, **: *p* ≤ 0.01, ***: *p* ≤ 0.001 and ****: *p* ≤ 0.0001.

## 3. Results

### 3.1. The EMT was Associated with A Loss of UPF1 Expression as well as GABARAPL1 Overexpression

Recent data showed that the key NMD factor *UPF1* is downregulated during the epithelial to mesenchymal transition (EMT) [21]. First, we confirmed these data in a cohort (*n* = 42) of non-small cell lung carcinomas (NSCLCs) in which the tissue samples were sorted according to their Vimentin status: EMT+ tumors that exhibit high percentages of malignant cells expressing this biomarker and EMT− tumors with a complete absence of expression for this protein by the cancer cells [22]. As shown in Figure 1 (bottom panel), EMT+ tumors presented lower levels of *UPF1* compared to their EMT− counterparts. Furthermore, according to recent data obtained in our laboratory, we also showed the overexpression of the ATG8 family member *GABARAPL1* in the case of the EMT (Figure 1, upper panel) (Jacquet et al., manuscript accepted biology-1328521). Even if a global increase in *GABARAPL1* associated with a downregulation of *UPF1* was observed in EMT+ tumors, the high variability of protein levels that was observed between tissue specimens (patient heterogeneity) did not allow for a significant inverse correlation.

The regulation of the expression of *GABARAPL1*, as well as the other members of the ATG8 family, remains unclear. One of the main reasons is that most of the studies investigated their regulation at the translational or post-translational level, and very few studied their post-transcriptional regulation. We then wondered whether ATG8 members could be targeted by the NMD to induce their degradation. The A549 human lung cancer cell line was used to determine the expression of *UPF1* and *GABARAPL1* at both the mRNA and protein levels during the EMT since this cell line was previously described to undergo the EMT in response to TGFβ/TNFα treatment [23]. The analysis of *UPF1* and *GABARAPL1* expressions during the induction of the EMT by a TGFβ/TNFα treatment showed the downregulation of *UPF1* transcripts, as well as the ones of other NMD factors, after 15 h of treatment for *UPF1* and after 6 h for *UPF2*, *UPF3A*, *3B* and *SMG1* (Figure 2a). This decrease was then followed by a re-expression when treatment was prolonged with no significant decrease after 48 h of treatment. These results were also confirmed at the protein level for *UPF1* (Figure 2c,d), with a drastic decrease in *UPF1* protein level after 6 h of treatment. Interestingly, we observed an inverse profile for *GABARAPL1* transcripts, with a high expression of *GABARAPL1* mRNA after 6 h of treatment. When the treatment was extended, the *GABARAPL1* mRNA level remained higher than in the control condition, but with higher variability, leading to a non-significant increase (Figure 2b). These results suggested that the inhibition of NMD could promote a rapid increase in *GABARAPL1* mRNA levels following TGFβ/TNFα induction.

### 3.2. GABARAPL1 Transcript Increased upon NMD Inhibition

To assess whether the *ATG8* family member transcripts subject to NMD, we performed chemical inhibitions of the NMD using cycloheximide (CHX, a protein synthesis inhibitor), G418 (promoting PTC readthrough) or NMDi14 (disrupting SMG7–*UPF1* interactions), which are drugs that were shown to inhibit the NMD [24,25,26]. First, amongst the drugs that were used to inhibit the NMD, only the CHX induced a significant effect on the described NMD targets used as positive controls, with an increase in *SC35* and *ATF4* mRNA levels in the A549 cell line. Indeed, following 5 h of treatment, the *ATF4* mRNA levels presented a 7-fold induction and the *SC35* levels presented a 15-fold induction (Figure 3a). Similar results were obtained in a second cancer cell line, namely, the SiHa cells (Appendix A). Among the *ATG8* family transcripts, only *GABARAPL1* levels and, to a lesser extent, *LC3B* levels were increased upon CHX treatment. Indeed, the *GABARAPL1* mRNA levels were increased about seven- and fourfold, after the CHX treatment in A549 and SiHa cells, respectively. The increase in *LC3B* was less important with the threefold induction in A549 and about twofold in SiHa cells (Figure 3a and Appendix A). No effect was observed for the other *ATG8* family transcripts or endogenous NMD-insensitive transcripts, such as *SF3B5* and *ATG5*. The effect of the CHX treatment on the induction of *GABARAPL1* and *LC3B* transcripts appeared to be independent of an indirect transcriptional effect that is linked to *ATF4*, a known NMD target, since *ATG5*, which is a known transcriptional target of *ATF4*, like the *ATG8* family members, was not increased upon CHX treatment. The treatment using NMDi14 for 24 h did not induce any significant effect on the induction of either *SC35* or *ATF4* transcripts, which are two described NMD targets, demonstrating an inefficient NMD inhibition. Moreover, the treatment of the cells with NMDi14 caused important cell death and a proliferation inhibition, both in SiHa and A549 cells, as well as in U2OS cells (data not shown). The treatment using G418, which can enhance a stop codon readthrough by the ribosome and is therefore used to inhibit the recruitment of NMD factors on transcripts harboring PTCs, had only a minor effect on the *ATF4*, *SC35* and *GABARAPL1* mRNA levels in SiHa cells and showed no effect on those same targets in A549 cells. Furthermore, the co-treatment of cells with both G418 and NMDi14 did not show any further increase compared to G418 alone.

To examine whether NMD inhibition could affect the subcellular localization, as well as the expression of *GABARAPL1* mRNAs, we performed RNA-FISH using specific fluorescent probes targeting the *GABARAPL1* mRNA on A549 cells that were treated, or not, with CHX. In both conditions, the *GABARAPL1* mRNAs were observed as punctate dots in the cytoplasm (Figure 3b). We observed a significant increase in *GABARAPL1* mRNA dots in CHX-treated cells (about 14 dots per cell) compared to untreated cells (about 4 dots per cell) (Figure 3c). Therefore, the RNA-FISH experiments confirmed an increase in *GABARAPL1* transcripts upon CHX treatment.

To inhibit the NMD with a higher specificity than with CHX, we used siRNA-mediated inhibition of the NMD by targeting the transcripts of two core NMD factors, *UPF1* and SMG1, alone or in combination. The inhibition of the NMD by targeting these two factors was previously described to lead to NMD inhibition [27] and seemed to be efficient since the levels of the positive controls *ATF4* and *SC35* were increased following *UPF1* and *SMG1* knockdown in both A549 and SiHa cells, with greater effects observed when using both siRNAs (Figure 4a and Appendix A). According to the results obtained with the CHX treatment, the *GABARAPL1* mRNA levels were increased when NMD was inhibited by *siUPF1* and *siSMG1*, with, once again, a greater effect obtained when used in combination in both A549 and SiHa cells (Figure 4a and Appendix A). This increase in *GABARAPL1* mRNA level upon siRNA-mediated NMD inhibition was also observed in the glioblastoma cells U87 (Appendix A). However, no significant increase in *LC3B* mRNAs was observed following siRNA transfection, suggesting that the effect observed for this gene upon CHX treatment was not due to a direct NMD inhibition and was probably due to an off-target effect of the CHX. The increase in *GABARAPL1* was then confirmed at the protein level using Western blotting (Figure 4b), with a significant increase of *GABARAPL1* protein levels after *siUPF1* and *siSMG1* co-transfection (Figure 4c).

### 3.3. UPF1 Was Associated with GABARAPL1 mRNAs

To confirm our above results and demonstrate a direct targeting of *GABARAPL1* transcripts by the NMD, we analyzed the association of the *UPF1* protein with different *ATG8* transcripts. To do so, RNA-IPs were performed using either an anti-*UPF1* antibody (RIP-*UPF1*) or a control IgG (RIP-CTL). The specificity and efficiency of the *UPF1* recovery were assessed using Western blotting (Figure 5a) and *UPF1*-bound mRNAs were analyzed using RT-qPCR (Figure 5b). A 10% input was used to normalize the amount of proteins used in the experiment. As expected, described endogenous NMD targets (*ATF4* and *SC35*), used as positive controls, were indeed significantly enriched in the RIP-*UPF1* compared to the RIP-CTL, while the mRNAs that were not targeted by the NMD (*H3B2* and *SF3B5*) and used as negative controls were undetected in both the RIP-CTL and the RIP-*UPF1*. Regarding the *ATG8* transcripts, the *GABARAPL1* and *LC3B* mRNAs were detected in the RIP-*UPF1* with a fourfold enrichment, but no association of *UPF1* with *GABARAP* mRNAs was detected. Even though *LC3B* transcripts were detected in the RIP-*UPF1*, we previously showed in Figure 4 that this transcript was not increased in cells in which the NMD was inhibited, suggesting that *LC3B* mRNAs were not targeted by the NMD. Taken together, our RIP data indicated that *UPF1* did not associate with all *ATG8* transcripts and demonstrated that *GABARAPL1* mRNAs were specifically recognized and targeted by the NMD.

### 3.4. The 3′UTR of GABARAPL1 mRNAs Contributed to Its Degradation by The NMD

The *GABARAP* family members presented a strong identity part of their coding regions but differed in the length of their *3′UTR*, with a long *3′UTR* for *GABARAPL1* (1276 bp) and *LC3B* (1671 bp) compared to *GABARAP* (433 bp) (Figure 6a). When we looked at the sequences of the *ATG8* mRNAs, it was surprising to see that *GABARAPL1* could be targeted by the NMD, while *GABARAP* could not since those transcripts were very similar, as shown on the scheme in Figure 6a. Indeed, their ORFs shared 79% identity at the mRNA level and 90% identity at the protein level. The main difference between these two transcripts lay in their *3′UTR*s, which was longer for *GABARAPL1* than the one of *GABARAP* (1276 nt vs. 433 nt, respectively) and they shared no identity. Since it was admitted that the NMD could target physiological transcripts harboring long *3′UTR*s (>1000 nucleotides), we proposed the hypothesis that the *GABARAPL1* mRNA was subject to degradation by the NMD via its long *3′UTR*. To test this hypothesis, the sequence of the *3′UTR* of *GABARAPL1* was cloned downstream of the ORF of *GABARAP* in a *pGFP*-*C1* plasmid, and the expressions of the fusion transcripts were analyzed at both the mRNA and protein levels. As shown in Figure 6c, the addition of the *3′UTR* of *GABARAPL1* seemed to decrease the expression of the fusion protein *GFP*-*GABARAP* since we observed fewer fluorescent cells expressing the fusion protein. This effect was confirmed using Western blotting, with a decrease in the expression of the fusion protein *GFP*-*GABARAP* (Figure 6d,e). To confirm that this decrease of the *GFP*-*GABARAP* protein levels was due to a decrease in the mRNA levels, the expression of the *GFP-GP* + *3′UTR* GL1 transcript was assessed using RT-qPCR (Figure 6b). The results showed that, indeed, the addition of the *3′UTR* of *GABARAPL1* downstream the ORF of *GABARAP* decreased the levels of the fusion transcript, thus confirming our hypothesis that *GABARAPL1* is targeted by the NMD via its long *3′UTR* since it can alone destabilize the *GABARAP* transcript, which was not initially targeted by the NMD.

### 3.5. eIF4A3 Regulated GABARAPL1 mRNA Degradation via Its 3′UTR

To decipher how the *GABARAPL1* mRNA could be targeted to NMD degradation via its long *3′UTR*, proteins bound to the *3′UTR*s of *GABARAP*, *GABARAPL1* and *LC3B* were analyzed using the AURA 2.5.2 database (Figure 7) [28]. We observed that *UPF1* was found to bind the *3′UTR* of *GABARAPL1*, as well as the one of *LC3B*, and no *UPF1* binding site was found on the *GABARAP* transcript, confirming our RIP-*UPF1* results. Interestingly, we observed four *eIF4A3* binding sites on the *3′UTR* of the *GABARAPL1* transcript that overlapped with the *UPF1* binding sites, but none on the *LC3B 3′UTR*. Therefore, we hypothesized that *eIF4A3*, together with *UPF1*, could regulate the degradation of transcripts via their long *3′UTR*. This model could explain why *UPF1* could bind the *3′UTR* of *LC3B* without inducing its degradation since it was not bound by *eIF4A3*.

To confirm the impact of *eIF4A3* on the levels of *GABARAPL1* transcripts, *eIF4A3* was inhibited using specific siRNAs and the levels of ATG8 transcripts were analyzed using RT-qPCR in A549 cells (Figure 8). As expected, increased mRNA levels of *GABARAPL1*, *ATF4* and SC35 were detected in the *eIF4A3*-depleted cells (Figure 8a), thus confirming its role in *GABARAPL1* mRNA regulation. The same effect was observed in a second cell line, SiHa cells, with a significant increase in *GABARAPL1* mRNA levels following the *sieIF4A3* treatment (Appendix A). Furthermore, *eIF4A3* inhibition did not lead to a significant increase in *LC3B* mRNA, suggesting that *eIF4A3* has a key role in the degradation of transcripts binding *UPF1* in their *3′UTR*. The effect of *eIF4A3* inhibition on *GABARAPL1* expression was then confirmed at the protein level using Western blotting (Figure 8c), and quantitative analysis of protein levels using Western blotting confirmed the elevated levels of *GABARAPL1* proteins in *eIF4A3*-depleted cells (Figure 8d). Altogether, these results suggest that the mechanism by which the NMD targeted *GABARAPL1* mRNA occurred through the association of both *UPF1* and *eIF4A3* on its *3′UTR*.

## 4. Discussion

In our study, we decided to investigate the regulation of autophagy genes at the post-transcriptional level since most of the precedent studies focused on transcription or post-translational levels. To do so, we concentrated our efforts on the key pathway involved in post-transcriptional regulation, which is the NMD, and we decided to use lung cancer as a pathology model since it was previously demonstrated that the NMD was decreased in this cancer and that *GABARAPL1*, an autophagy gene, was increased. Our results indeed confirmed that the levels of the key NMD factor *UPF1* were downregulated in patients presenting with lung cancers with high levels of VIMENTIN, a key marker of the EMT. The EMT was described as being involved in tumor progression, and TGFβ signaling was shown to play an important role in this process. In vitro, the downregulation of *UPF1* during the EMT in a lung adenocarcinoma cell line was previously described by Cao et al. [21], where the authors showed that the inhibition of NMD led to the induction of the EMT and that the overexpression of NMD factors led to the inhibition of the EMT. The authors also suggested that this regulation of the EMT by the NMD occurred through the targeting of factors involved in the TGFβ signaling pathway, suggesting an important role of the NMD pathway in the induction of the EMT. Interestingly, we observed that the decreased levels of *UPF1* during the EMT were associated with an increase in *GABARAPL1* expression. The ATG8 family member *GABARAPL1* was shown to be required for the degradation of damaged organelles or protein aggregates via selective autophagy [29], and recent results obtained in our laboratory showed a role of *GABARAPL1* during the EMT (Jacquet et al. manuscript under review in Biology). Even though the regulation of *GABARAPL1* during the EMT occurs, in part, through a transcriptional level, the mechanisms behind its regulation are not fully understood. Our results demonstrated that NMD factors were expressed at low levels during the EMT and that this expression was inversely correlated with the levels of *GABARAPL1* in lung cancer models, suggesting that *GABARAPL1* could also be regulated during the EMT at the post-transcriptional level by the NMD pathway as an interplay between mRNA degradation pathway and autophagy. Moreover, crosstalk between autophagy and selective mRNA degradation was previously described in yeast by Makino and collaborators [30], demonstrating that autophagy could selectively target and degrade mRNAs.

To confirm that *GABARAPL1* was directly targeted by the NMD, this pathway was inhibited by both chemical inhibitors or siRNAs. These inhibitions led to the increase in *GABARAPL1*, both at the mRNA and protein levels, confirming that the NMD could regulate autophagy transcripts, as suggested by Wengrod et al. [31]. We also confirmed the overexpression of *ATF4* mRNA upon NMD inhibition. However, our results differed from the ones of Wengrod and collaborators since we did not observe any increase in *ATG5* mRNA levels. Wengrod and colleagues also showed that the inhibition of the NMD led to the induction of autophagy in an indirect manner through the targeting of *ATF4*, which is a transcription factor targeted by the NMD [32]. Since this transcription factor is known to activate the transcription of many autophagy-related genes, including *ATG5* and several *ATG8* family members, these data could explain the differences between our data and the ones from Wengrod and collaborators. This was confirmed by the fact that we did not observe any increase in other ATG8 family members, even though several of them were previously shown to be regulated at the transcriptional level by *ATF4* [4,5,33]. Therefore, our data indicated that the *GABARAPL1* upregulation was not occurring through an indirect *ATF4* regulation but through direct targeting of its transcripts by the NMD pathway. However, transcriptional regulation of *ATG8* members’ expressions by *ATF4* following NMD inhibition is not excluded since we observed a slight but non-significant increase of *ATG5* mRNA levels following NMD inhibition by siRNAs, but we might have analyzed the transcripts’ expressions too early following NMD inhibition to observe this transcriptional regulatory effect.

Then, our RIP-UPF1 results showed that *UPF1* did bind the *GABARAPL1* mRNA, as well as the one of *LC3B*. The mechanism of *GABARAPL1* mRNA recognition by the NMD pathway remained quite unclear since this transcript contained no known premature termination codon. However, it was previously proposed in a study analyzing the mechanism by which long *3′UTR* mRNAs elicit or escape the NMD that the *3′UTR* of *GABARAPL1* could promote NMD degradation, leading to the hypothesis of targeting through its *3′UTR* [34]. We then cloned the *GABARAPL1 3′UTR* downstream of the coding sequence of *GABARAP* and confirmed that *GABARAPL1* was targeted via its *3′UTR*, suggesting a degradation by an EJC-independent NMD, which was previously suggested for transcripts harboring a long *3′UTR* (>1000 nt) [16,35,36]. However, the size of the *3′UTR*, as well as its binding by *UPF1*, did not seem to be sufficient to induce NMD degradation since numerous transcripts harboring long *3′UTR* were never described as NMD targets, and that *LC3B*, which also presents a long *3′UTR*, did not seem to be degraded by the NMD, even if this transcript was found to be enriched upon *UPF1* RNA-IP.

Using the Atlas of UTR Regulatory Activity (AURA) data, we found that the EJC factor *eIF4A3* bound the *3′UTR* of *GABARAPL1* but not the ones of *LC3B* and *GABARAP*. Previous studies indicated that *eIF4A3* occupancy was variable along mRNAs with both canonical and noncanonical EJC binding sites [11,37]. Interestingly, *eIF4A3*-binding sites were located in proximity to *UPF1*-binding sites on the *3′UTR* of *GABARAPL1*. Therefore, we hypothesized that *eIF4A3*, together with *UPF1*, could regulate the degradation of transcripts via their long *3′UTR*s. This model could explain why *UPF1* could bind the *3′UTR* of *LC3B* without inducing its degradation since it was not bound by *eIF4A3*.

To further decipher the features that could lead transcripts harboring long *3′UTR* being degraded by the NMD, we then analyzed the proteins bound to *GABARAPL1* mRNA, as well as other NMD targets, in proximity to *UPF1*. This analysis confirmed that *UPF1* could bind both *GABARAPL1* and *LC3B 3′UTR*s, as well as GADD45B and TBL2 *3′UTR*s, which are two transcripts that were previously described to be targeted by the NMD through their *3′UTR*s [38]. The most important outcome of this analysis was the systematic presence of *eIF4A3* in proximity to *UPF1* on *GABARAPL1*, as well as *GADD45B* and *TBL2*, but there was an absence of *LC3B 3′UTR*. This result led us to think that the discrimination of NMD targets between all transcripts that were bound by *UPF1* in their *3′UTR* could depend on the presence of *eIF4A3*. Moreover, the same protein environment was observed in proximity to *UPF1* for these four transcripts, with the presence of proteins such as *RBM47*, *LIN28*, or *AGO*. For the latter, it was suggested that *UPF1* could play a role in the argonaute pathway and the degradation of transcripts by miRNAs, which is a mechanism that then may be involved in the degradation of transcripts that bind *UPF1* in their *3′UTR*s [39]. However, even if *LC3B* did not seem to be degraded by the NMD since it did not respond to NMD inhibition from siRNA targeting, it would be interesting to further analyze the role of *UPF1* in the regulation of this transcript since it responded to CHX treatment and was enriched upon *UPF1* RNA-IP. However, the impact of *UPF1* binding on the *LC3B* mRNA is still unclear, but we hypothesize that it may have a role in the subcellular localization of *LC3B* since *UPF1* was previously described to present non-NMD roles, such as transport and localization of many cellular transcripts, but further experiments will be needed to confirm this hypothesis.

Altogether, our results led us to propose a model in which the *3′UTR*-dependent NMD may be an important mechanism for the induction of autophagy during the EMT via the regulation of the levels of the *GABARAPL1* transcript (Figure 1). However, we are aware that further studies are required to determine whether the modulation of the NMD pathway during the EMT could contribute to the induction of selective autophagy mediated by *GABARAPL1* and its consequences on tumor progression.

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
