# Peer review of "The NMD Pathway Regulates GABARAPL1 mRNA during the EMT"

_biomedicines, 2021, doi:10.3390/biomedicines9101302_

Round 1

Reviewer 1 Report

Dear authors of the manuscript:

The NMD pathway differentially regulates ATG8 mRNAs during EMT

I see the problem describing the complexity of this mechanism. I would recommend to spend one ore two more sentences in the beginning of Introduction and the discussion to provide the reader more about the overall context. During reviewing I needed to read additional publications in the field to find the context explanation I was looking for e.g. Cao et al 2017, Makino et al. nature communication 2021 or Wengrod 2013.

Please simplify  the discussion between line 476 and 506 in a way to bridge the results to the final conclusion and the description of Scheme1. to provide the reader a better way to understand the new findings ant the take home message.

Author Response

Response to Reviewer 1 Comments

Comments and Suggestions for Authors

Dear authors of the manuscript:

The NMD pathway differentially regulates ATG8 mRNAs during EMT

Point 1: I see the problem describing the complexity of this mechanism. I would recommend to spend one ore two more sentences in the beginning of Introduction and the discussion to provide the reader more about the overall context. During reviewing I needed to read additional publications in the field to find the context explanation I was looking for e.g. Cao et al 2017, Makino et al. nature communication 2021 or Wengrod 2013.

First, we would like to thank reviewer 1 for his positive reviewing.

The publications of Cao et al. (2017) and Wengrod et al. (2013) were cited in the original version, we nevertheless modified the manuscript to provide a more general context and a better explanation of these complex mechanisms of NMD and EMT (see our modifications highlighted in red in the manuscript). We also added the conclusions of the publication of Makino et al. (2021) in the discussion and we modified the introduction to give more information to the reader.

Please simplify the discussion between line 476 and 506 in a way to bridge the results to the final conclusion and the description of Scheme1. to provide the reader a better way to understand the new findings ant the take home message.

According to your suggestion, we modified this part of the discussion to simplify it and concentrate on our new findings and future outcomes.

Reviewer 2 Report

In this manuscript entitled “The NMD pathway differentially regulates ATG8 mRNAs during EMT” Baudu and collegues investigated the role of NMD pathway in the regulation of GABARAP, GABARAPL1, LC3B autophagy related proteins. The authors provide experimental evidence that the NMD factor UPF1 in collaboration with the eif4A3 protein binds the 3’ UTR of GABARAPL1 mRNA mediating its degradation. Based on these data, they report that EMT-driven downregulation of UPF1 leads to higher GABARAPL1 mRNA levels.

The post-transcriptional regulation of autophagy genes is very little studied and therefore this is an interesting study that provides novel data about the EMT/NMD/autophagy crosstalk. However, the vast majority of the conclusions has been based on experimental work conducted only in one cell line (A459) and this is a major weakness in this study.

Comments:

  • The experimental data should be validated by including additional cell lines (at least one more) in the analysis.  
  • A meta-analysis of TCGA lung adenocarcinoma data could validate the reported inverse correlation between vimentin and UPF1 expression levels (EMT cluster). An Inverse correlation should also exist between UPF1 and GABARAPL1 levels.
  • Figure 8: Silencing of eIF4A3 levels is nor presented (western blot or qRT-PCR analysis).
  • Since the most solid experimental data have been produced for GABARAPL1, The title of the manuscript should focus on GABARAPL1 gene and not generally on “ATG8 mRNAs”.

Round 2

Reviewer 2 Report

I consider that it is important the RNA-immunoprecipitation experiment to be performed at least in a second cell line. This experiment is clearly missing from the study. However I think that this study provides a first line of experimental evidence that UPF1 in collaboration with the eif4A3 protein binds the 3’ UTR of GABARAPL1 mRNA mediating its degradation. In this context, I would suggest the publication of this study in its current form.